# Gender, Self-Efficacy, and Academic Performance: Evidence in Business Education Program

**DOI:** 10.3390/bs15050563

**Published:** 2025-04-22

**Authors:** Ana Moraga-Pumarino, Sonia Salvo-Garrido, Vesnia Ortiz-Cea

**Affiliations:** 1Departamento de Administración y Economía, Universidad de La Frontera, Temuco 4780000, Chile; vesnia.ortiz@ufrontera.cl; 2Departamento de Matemática y Estadística, Universidad de La Frontera, Temuco 4780000, Chile; sonia.salvo@ufrontera.cl

**Keywords:** self-efficacy, business education, gender, higher education

## Abstract

Self-efficacy, the conviction in one’s ability to succeed in particular tasks, is crucial in academic performance and professional development, especially in higher education programs. Although it has been widely studied in STEM disciplines, research on gender differences in self-efficacy in business education is scarce, especially in the Chilean context. This study aims to fill this gap by examining self-efficacy beliefs and their association with academic performance among business students at a Chilean university. Using a quantitative, cross-sectional design, data were collected from 239 students via the validated Academic Self-Efficacy Scale (ESAA). Confirmatory factor analysis confirmed the scale’s psychometric robustness in measuring situational and personal efficacy beliefs. No significant associations were found between self-efficacy and academic performance. However, gender differences emerged in personal efficacy beliefs, with men reporting higher levels than women, while no differences were found in situational efficacy. These findings reinforce the multidimensional nature of academic self-efficacy and reveal persistent gender disparities in how students perceive their individual academic capacities. This asymmetry may limit women’s engagement, confidence, and long-term professional development in male-dominated fields such as business. The results point to the need for gender-sensitive educational strategies that intentionally foster personal efficacy beliefs among female students, thereby promoting more equitable academic experiences and professional trajectories.

## 1. Introduction

Self-efficacy, defined as the belief in one’s ability to organize and execute actions to achieve goals ([5], [6], [7]), is broadly linked to academic performance and student persistence ([18]; [21]; [32]; [52]; [80]). According to [8] ([8]), self-efficacy directly affects behavior and influences other determinants, such as goals, expectations, and perceptions of the environment. This idea is essential for students’ adjustment to academic demands, enhancing motivation, perseverance, and academic success. It is crucial for personality development and enhancing learning ([24]; [72]; [83]). Students with strong self-efficacy exhibit greater intrinsic motivation, whereas those with low self-efficacy depend more on external motivations ([50]). In addition, it is related to the ability to face academic challenges progressively ([39]), influenced by personal achievement, social comparison, and external support ([3]; [10]; [33]).

At the university level, self-efficacy is a significant predictor of academic performance ([36]; [44]; [46]; [70]; [76]) and facilitates stress management and adaptation to the demands of education programs ([13]; [26]; [27]; [28]). According to [89] ([89]), self-efficacy mitigates stress as a predictor of academic performance, significantly contributing to surmounting obstacles and student perseverance. [91] ([91]) asserts that tolerance levels to academic stress vary among student profiles, highlighting that students with high self-efficacy view stress as a challenge that fosters their commitment and satisfaction, thereby adopting positive attitudes towards academic demands. In this vein, [51] ([51]) emphasize the need for universities to promote environments that reduce stress and strengthen self-efficacy through technological tools and psychological support programs.

Self-efficacy also indirectly affects academic performance through motivation and persistence ([4]; [13]; [28]; [42]; [92]); then, students with higher self-efficacy are generally more motivated to confront academic challenges and persist in their studies. This is particularly relevant during the first years of university ([13]; [53]; [87]; [89]). Their development can be enhanced through a variety of learning opportunities ([25]; [42]) since, as indicated by [9] ([9]), self-efficacy beliefs are flexible, subject to development, and shaped by multiple resources.

Self-efficacy has a positive impact on student engagement, suggesting that strengthening it can significantly improve participation in educational activities ([19]). Students with higher self-efficacy seek active feedback and share ideas without worrying about risks like public exposure ([29]). However, high self-efficacy levels are not always beneficial. [12] ([12]) warns that these beliefs can result in performance disparities, even among individuals with comparable skills, and excessive levels of self-efficacy may be detrimental in certain settings ([78]; [84]). Moreover, although there is an overall positive correlation between academic self-efficacy and academic performance, [30] ([30]), [43] ([43]), [65] ([65]), [68] ([68]) and [73] ([73]) note that its effect depends on the context and individual factors.

In the Chilean context, studies are limited, but existing evidence confirms the positive correlation between self-efficacy and academic performance. According to [34] ([34]), self-efficacy expectations are associated with improved academic outcomes, a correlation observable across various disciplines. For their part, [56] ([56]) identified a positive connection between the use of learning strategies and self-efficacy, with moderate levels, emphasizing that a heightened sense of self-efficacy improves both motivation and autonomy in the learning process. Consistent with other studies, [82] ([82]) reported that academic self-efficacy is positively related to academic performance and depends on the area of training, which is higher among basic science students and lower in teaching programs. Furthermore, [56] ([56]) identified that first-year students present higher levels of self-efficacy than second-year students, suggesting a possible decline as they progress in their studies. Paradoxically, [82] ([82]) concluded that self-efficacy can increase at advanced stages of the curriculum.

In light of the multidimensional nature of academic self-efficacy, this study employs the ESAA (Escala de Autoeficacia Académica), a validated instrument that distinguishes between two core components of efficacy beliefs ([71]). The first refers to situational efficacy, or students’ perceptions of how external academic conditions (e.g., teaching methods, institutional support, learning environment) influence their performance. The second is personal efficacy, which encompasses students’ internal beliefs regarding their ability to organize, manage, and succeed in academic tasks. This distinction is fundamental to understanding the mechanisms through which self-efficacy influences academic outcomes and provides a more nuanced analytical framework for evaluating individual and contextual contributions to student achievement.

### 1.1. Gender and Self-Efficacy

Despite its relevance, self-efficacy presents significant gender differences in its development and manifestation. [38] ([38]) conducted a meta-analysis of 247 studies, identifying a slight advantage in male academic self-efficacy, especially in areas such as mathematics, computer science, and social sciences. In contrast, women tend to excel in disciplines such as language arts and the arts. [38] ([38]) and [53] ([53]) also noted a general trend toward higher levels of self-efficacy in men, although not in all study domains, highlighting that the specific subject matter serves as a key moderator in this relationship. These findings indicate that women exhibit greater self-efficacy in verbal domains, including language and literature, whereas men demonstrate superiority in technical and scientific fields. Conversely, [88] ([88]) highlight that gender differences influence the academic self-concept, affecting performance, motivation and execution.

Gender differences in self-efficacy vary by subject area and context. [88] ([88]) point out that gender stereotypes impair women’s mathematical and scientific self-concept, limiting their participation in STEM fields while they excel in verbal domains. These disparities, [88] ([88]), underscore the need for equitable educational strategies that narrow gender gaps and promote female self-efficacy in male-dominated disciplines. Nonetheless, these disparities are neither uniform nor consistent across all places and periods. For example, in online learning environments, women show higher levels of self-efficacy than men, especially in countries like the United States ([42]). In addition, studies indicate that Latina women in higher education experience higher stress levels than their male counterparts, which may negatively impact their perceived self-efficacy ([17]).

The impact of gender on self-efficacy transcends academia and is influenced by cultural factors, gender roles, and family attitudes ([38]; [46]). Liberal attitudes to gender roles are associated with higher levels of self-efficacy, with the exception of white men, for whom these perspectives seem to exert less influence ([53]). In general, men tend to report higher self-efficacy in complex tasks, such as the use of advanced technologies ([46]), while women experience greater anxiety in academic settings with prominent traditional gender roles, which negatively affects their self-efficacy ([59]). These findings reinforce the need for context-specific interventions that promote self-efficacy in both genders.

### 1.2. Business Education Programs and Self-Efficacy

In business studies, self-efficacy is essential due to the critical competencies needed in this area, including leadership, strategic decision-making, and complex problem-solving ([2]; [22]; [23]; [41]; [47]; [55]). Studies have shown that students with high self-efficacy not only achieve better academic outcomes but also show a greater inclination toward entrepreneurship and professional success ([11]; [14]). Regarding classroom participation, [86] ([86]) stress that self-efficacy directly affects student engagement, highlighting the importance of cultivating this skill in the educational process. In addition, [1] ([1]) and [47] ([47]) highlight its essential role in promoting entrepreneurial intent, a particularly pertinent element in business education. However, its impact is not limited to the academic environment, as it also contributes significantly to preparation for the labor market ([66]). According to H. [90] ([90]), MBA students with high self-efficacy show greater confidence and understanding, which are essential for securing employment or starting a business, as well as fostering positive work attitudes and career planning skills. Consistent with this, [47] ([47]) reported that accounting students with high academic self-efficacy obtained better academic outcomes, a conclusion corroborated by [14] ([14]), who determined that confidence in one’s own abilities increases the likelihood of academic success. For their part, [69] ([69]) note that self-efficacy can also strengthen students’ communication skills.

Notwithstanding these advances, a wide gap persists in understanding how gender differences affect business students specifically. Unlike STEM disciplines, where gender differences and self-efficacy have been studied extensively, this area has received less attention in business education. A study in this regard is that of [86] ([86]), which identified higher levels of self-efficacy in men than women. However, these results have not been replicated in other business settings or sub-disciplines, complicating the generalization of the findings. This lack of evidence is particularly concerning given that fields such as accounting, management, and entrepreneurship remain male-dominated in many contexts, perpetuating invisible barriers to women’s participation and professional development. In the Chilean context, business education programs are characterized by a comprehensive curriculum that combines subjects such as accounting, economics, finance, strategic management, and entrepreneurship alongside active methodologies and practical experiences aimed at developing key competencies such as leadership and decision-making. Within this demanding and competitive educational environment, academic self-efficacy emerges as a critical factor for both academic achievement and students’ professional projection. Understanding how gender differences manifest in this variable is essential for designing more inclusive and effective pedagogical strategies. As suggested by ([48]), interventions specifically designed to strengthen self-efficacy—particularly among women—may have a significant impact on their long-term academic and professional success. Such interventions may include mentoring, leadership training, or self-regulated learning programs, which have been shown to positively influence students’ academic self-efficacy ([36]; [60]).

This study aims to contribute to the literature by addressing this gap through the analysis of the relationship between academic self-efficacy and academic performance among business students, with a particular focus on gender differences. The proposed hypotheses are: (H1) there is a positive relationship between academic self-efficacy and academic performance; (H2) there are significant differences in self-efficacy levels according to gender; and (H3) the relationship between academic self-efficacy and academic performance varies as a function of gender.

## 2. Materials and Methods

The study, based on a positivist approach with a quantitative, cross-sectional, descriptive design and inference ([20]), used the Academic Self-Efficacy Scale (ESSA), validated in the Peruvian university population ([71]) and grounded in Bandura’s theory of self-efficacy ([6]). The ESAA consists of 28 items distributed into two main factors: (1) situational efficacy beliefs (8 items), which assess students’ perceptions of how external factors—such as institutional resources, teaching methodologies, or academic infrastructure—impact their performance. An example item is: “My academic performance depends on the support I receive at the university”. (2) personal efficacy beliefs (20 items), which reflect students’ internal beliefs about their ability to succeed in academic tasks. Example items include: “I am confident I can reach my academic goals” and “I have the ability to overcome academic difficulties”. The scale uses a 5-point Likert-type format, where 1 = “Never”, 2 = “Almost never”, 3 = “Sometimes”, 4 = “Almost always”, and 5 = “Always”. Higher scores indicate stronger academic self-efficacy beliefs. The instrument demonstrated strong psychometric properties, including high internal consistency (α = 0.877) and sampling adequacy (KMO = 0.952).

The study population comprised all business students at a Chilean public university enrolled from the second year onwards (N = 525). The entire population was invited to participate, resulting in a final sample of 342 students (65% of the population). The main variables were gender, academic performance measured as self-reported GPA (grade point average; scale 1–7), and academic self-efficacy measured with the ESAA ([71]).

Procedure: Students were contacted via e-mail and in person in the classrooms, inviting them to participate voluntarily and anonymously. Data were collected using the Question Pro platform, including an informed consent form, the ESAA, and sociodemographic and academic characterization questions, including academic performance and self-reported gender.

Data analysis: A confirmatory factor analysis (CFA) with ULSMV estimation was performed for ordinal variables, assessing reliability using Cronbach’s α and ω coefficient (acceptable values: 0.70–0.90) ([15]). The CFI and TLI (>0.90) and RMSEA (<0.08) indices determined the fit of the model. The relationship between self-efficacy and academic performance was also explored (Pearson or Spearman coefficients according to normality) and gender differences using Student’s *t*-tests or Mann-Whitney tests. In addition to treating academic performance as a continuous variable, the sample was also classified into three performance groups based on empirical terciles. This complementary categorization enabled a more nuanced comparison across performance levels and facilitated the identification of specific subgroup differences. This approach is consistent with recommendations in the literature advocating for multi-level groupings to capture variation in student outcomes more effectively than binary splits ([85]). The empirical cutoff points derived from the data were as follows: low performance (≤5.4), medium performance (>5.4 and ≤5.6), and high performance (>5.6). This classification provided a balanced distribution of participants and helped detect patterns in self-efficacy that may vary across performance levels. One-way and factorial ANOVA tests were performed to analyze differences in academic self-efficacy across gender and academic performance levels. The decision to apply parametric or non-parametric procedures was based on the assessment of the normality assumption. Specifically, the Shapiro–Wilk test was used to evaluate the distribution of the dependent variables within groups ([77]). When the assumption of normality was met, parametric ANOVA was used; otherwise, the non-parametric alternative was applied. For the one-way ANOVA, the Kruskal–Wallis test was used in case of non-normal distributions ([54]). For the factorial design involving two independent variables (gender and GPA categories), the Scheirer–Ray–Hare test was used as a non-parametric alternative to the two-way ANOVA ([75]). Effect sizes were reported using eta squared (η^2^) or epsilon squared (ε^2^) as appropriate. Violin plots were used, when relevant, to visualize significant group differences in self-efficacy ([79]). 

Software: Analyses were performed with Mplus v.8.4 ([58]) and JAMOVI v.1.8.1 ([40]). The latter is noted for generating visually informative data distribution graphs ([35]).

## 3. Results

Initially, 242 students participated in the study, and 98% reported their gender as male or female; therefore, those reporting another gender were excluded due to the constraints in conducting comparative analyses on such small groups. Hereafter, the analysis will focus on the male and female genders exclusively. The final sample comprised 239 students, representing 43% of the eligible population. The demographic profile of the sample shows a balanced gender distribution (male: 48.1% and female: 51.9%). The average age was 21.6 years (SD = 2.33), and most of the sample (90%) entered the program by regular admission. The students are homogeneously distributed in different curriculum levels, and the average self-reported academic performance was 5.46 (SD = 0.363). Table 1 shows further features of the sample.

### 3.1. Confirmatory Factor Analysis (CFA) ESSA Scale

A CFA was performed with all the data obtained. The scale presented good psychometric properties in the Chilean population. The proposed theoretical model with two correlated factors presented a good fit to the data: CFI = 0.945; TLI = 0.941; RMSEA = 0.052 (CI90% = 0.045 0.06), SRMR (standardized root means square residual) 0.067. The internal consistency indicators of the ESAA were Cronbach’s alpha = 0.846 and omega = 0.886, corroborating the results reported in the Peruvian population. Table 2 presents the adjusted model that yielded two factors, Factor 1 and Factor 2, as suggested by the original model, together with the measurement items, their standardized factor loadings, and corresponding standard errors. Factor loadings ranged between 0.325 and 0.711 for Factor 1 (situational efficacy beliefs) and between 0.315 and 0.865 for Factor 2 (personal efficacy beliefs), and all were statistically significant (p-v < 0.0001). The correlation between the two factors was −0.174, showing a negative and significant correlation.

The results (Table 3) reveal that Factor 1 scores are concentrated in the middle of the scale, suggesting a neutral or slightly middle-valued perception of situational efficacy beliefs (Mean = 3.01, SD = 0.592). In contrast, Factor 2 reflects a high perception, indicating that students positively value their personal efficacy beliefs (Mean = 3.97, SD = 0.539).

In addition, the coefficient of variation in both factors is less than 30%, which reinforces the representativity of the means concerning the data set. As for the distribution of the results, both skewness and kurtosis are within the acceptable range of ±1.5, suggesting an approximately normal distribution of the scores. This finding is complemented by the Kolmogorov-Smirnov test, which confirms the normality of the distribution in both factors (*p* > 0.05).

### 3.2. Academic Self-Efficacy and Performance

The students’ self-reported academic performance presented a mean of 5.46 (SD = 0.3629) on a scale from 1.0 to 7.0. A Pearson correlation was conducted between self-reported performance and each of the Academic Self-Efficacy factors, revealing no statistically significant associations for any of the factors (*p*-value > 0.10). As a complementary analysis, the performance groups previously defined (low, medium, and high) were used to examine potential differences in self-efficacy. Table 4 presents the descriptive statistics by performance group.

The assumption of normality was evaluated using the Shapiro–Wilk test. Results indicated deviations from a normal distribution for both situational efficacy beliefs (Factor 1; W = 0.986, *p* = 0.019) and personal efficacy beliefs (Factor 2; W = 0.979, *p* = 0.001). Given these violations, non-parametric Kruskal–Wallis tests were conducted to compare self-efficacy scores across the three GPA-based performance groups. For Factor 1, no statistically significant differences were found (χ^2^(2) = 0.429, *p* = 0.807). Similarly, for Factor 2, group differences were not significant (χ^2^(2) = 4.46, *p* = 0.108). These results indicate that academic performance level does not substantially affect students’ self-efficacy.

### 3.3. Academic Self-Efficacy by Gender

Comparisons between groups were analyzed to identify possible differences in students’ perceptions of academic self-efficacy in each factor on the scale based on gender. The sample included 115 male and 124 female students, both with a normal distribution, which justified the use of the Student’s *t*-test for independent samples in the analysis of differences in means. Table 5 shows the mean scores for each of the factors of the ESAA disaggregated by gender. For this analysis, the sum of the items of each factor was used instead of the score derived from the calculation. This decision was based on the fact that the total showed a high correlation (r = 0.98) with the score, facilitating a more straightforward and comprehensible interpretation of the results.

The results indicate no significant differences in self-efficacy perceptions related to Factor 1 between men and women (*p*-value > 0.10). This suggests that perceptions of self-efficacy regarding situational efficacy beliefs are highly similar between male and female students. These findings reinforce the conclusion that there are no significant differences between genders in this factor, which implies that the perception of the academic environment does not vary according to gender.

In contrast, for Factor 2, statistically significant differences were observed between genders, favoring male students (*p*-value < 0.001). The effect size, measured by Cohen’s d, is 0.5597, representing a medium effect. This value signifies that the disparity in personal efficacy beliefs is both statistically significant and relevant in practical terms, demonstrating a clear trend in favor of male students in their perceptions of self-efficacy in this factor.

Additionally, violin plots were generated to illustrate the differences in academic self-efficacy between genders. Figure 1 shows that the distribution of Factor 1 almost overlaps between men and women, visually confirming the lack of significant differences identified in the statistical analysis. In contrast, for Factor 2, the violin plot reveals a shifted distribution in favor of male students, highlighting a notable difference in self-efficacy perceptions. These findings, supported by both statistical analysis and illustrations, suggest that male students have significantly higher perceptions of personal efficacy beliefs compared to their female classmates.

### 3.4. Academic Self-Efficacy, Performance and Gender

A Pearson correlation analysis was first conducted to explore the linear relationship between self-reported academic performance and both academic self-efficacy factors, disaggregated by gender. No statistically significant associations were found for either factor (*p* > 0.10) in male or female subgroups. To further examine group-level differences, a non-parametric factorial ANOVA (Scheirer-Ray-Hare test) was conducted using gender and GPA-based performance categories. For situational efficacy beliefs (Factor 1), the analysis showed no significant main effects of academic performance (H = 0.089, *p* = 0.765) or gender (H = 0.007, *p* = 0.932), and no significant interaction between the two variables (H = 0.000, *p* = 0.990). In contrast, for personal efficacy beliefs (Factor 2), a significant main effect was found for gender (H = 15.29, *p* < 0.001, ε^2^ = 0.064), while academic performance (H = 0.28, *p* = 0.598) and the interaction term (H = 0.52, *p* = 0.470) were not statistically significant. These results indicate that personal efficacy beliefs differ significantly by gender, whereas academic performance level and the interaction between factors do not appear to influence self-efficacy in a meaningful way.

## 4. Discussion

This study evaluated the Academic Self-Efficacy Scale (ESAA) in a sample of Chilean business education students, affirming its robust psychometric properties, including internal consistency and factorial validity. The two-factor structure of the ESAA, preserved in this study, reflects a fundamental theoretical distinction. Situational efficacy beliefs refer to students’ perceptions of how external academic conditions—such as teaching resources, institutional infrastructure, or classroom climate—support or hinder their performance. In contrast, personal efficacy beliefs reflect students’ internal confidence in their ability to face academic challenges, persevere, and achieve success. This conceptual separation, grounded in Bandura’s social cognitive theory ([6]) and operationalized in the ESAA ([71]), is essential to understand the multidimensional nature of academic self-efficacy and to interpret the differentiated results between factors.

The validation of the scale preserved its two-factor structure in the study population, with a good fit to the data: CFI = 0.945; TLI = 0.941; RMSEA = 0.052 ([37]). These results reinforce its usefulness as a robust tool for measuring academic self-efficacy in Spanish-speaking settings, providing a solid foundation for future studies. The negative and moderated correlation (−0.174) between situational and personal efficacy factors highlights an inverse relationship between these dimensions of academic self-efficacy. This finding is consistent with [6] ([6]) assertion that efficacy beliefs are multifaceted and context-dependent. [71] ([71]) emphasized this theoretical distinction in the ESAA’s development, noting that situational efficacy is shaped by environmental factors, while personal efficacy relies on cumulative experiences and intrinsic resilience. The negative correlation may suggest that increased situational confidence in certain contexts could coincide with doubts about broader academic abilities.

The study did not confirm hypothesis H1. The analyses performed, including Pearson’s correlation and Student’s *t*-test, yielded no statistically significant associations between the factors of academic self-efficacy (situational efficacy beliefs and personal efficacy beliefs) and self-reported academic performance. This result was consistent whether performance was analyzed as a continuous variable or categorized into low, medium, and high GPA groups. This outcome aligns with the previous literature, which highlights the moderate and variable influence of self-efficacy on academic performance ([36]; [68]). The absence of significant relationships could be explained by the homogeneity of the sample in terms of academic perceptions and achievement levels, with a mean of 5.46 (SD = 0.3629) on a scale of 1.0 to 7.0. This relatively narrow score range could have constrained the variability needed to observe statistically significant relationships. The literature highlights that this relationship is not linear but rather is mediated by multiple factors, such as stress, intrinsic motivation, and institutional dynamics ([36]; [51]). [49] ([49]) suggest that self-efficacy and academic performance may interact reciprocally, generating a positive cycle where success reinforces confidence in one’s own abilities. However, this study found no evidence of such a cycle, which could be related to the specific characteristics of the study context.

The analysis by gender partially confirmed H2, identifying significant differences in the personal efficacy beliefs factor but not in situational efficacy beliefs. Specifically, men reported significantly higher levels of personal efficacy beliefs (*p*-value < 0.001), while no relevant differences were observed in situational efficacy beliefs (*p*-value > 0.10). The violin plots reinforce these observations, revealing a uniform distribution of situational efficacy beliefs but a clear shift in personal efficacy beliefs in favor of men. This finding suggests that, although men and women perceive the conditions of their academic environment similarly, there are differences in how they evaluate their own personal capabilities. The absence of significant differences in situational efficacy beliefs indicates that men and women have similar perceptions of the learning environment, which is in line with previous research highlighting homogeneity in the evaluation of academic contexts between genders ([29]; [36]). However, the disparities noted in personal efficacy beliefs expose a gender gap that broader social and cultural dynamics may influence. [49] ([49]) highlight that women face higher perceived barriers and tend to be more critical of their abilities than men, particularly in areas like business. The disparities observed in personal efficacy beliefs are consistent with [74] ([74]) and [89] ([89]), who argue that men tend to overestimate their abilities and past performance, whereas women tend to underestimate or evaluate them more accurately. Such a pattern is observed in the perception of one’s own competence and in the confidence to complete specific tasks ([81]). [38] ([38]) found a persistent advantage for men in academic self-efficacy, particularly in fields requiring leadership, analytical skills, and decision-making. Similarly, [86] ([86]) reported that male business students consistently exhibit higher confidence in their abilities, a critical factor in fostering academic success and engagement. These disparities are not limited to business education but are deeply rooted in systemic issues, as highlighted by [59] ([59]) and [46] ([46]), who emphasize the role of societal expectations and gender norms in shaping self-efficacy perceptions. [67] ([67]) argue that these gaps are exacerbated in contexts with entrenched cultural norms, limiting women’s potential in academic and professional domains. [6] ([6]) also notes that self-efficacy is influenced not only by individual experiences but by broader social and environmental factors. In this context, gender stereotypes—intertwined with societal expectations—play a central role in undermining women’s confidence and self-perception, placing them at a disadvantage. The findings also align with [16] ([16]), who highlight that men report greater confidence in leadership and decision-making tasks, which are essential in business settings. [38] ([38]) reinforces this view by noting that such disparities are sustained in environments where high self-efficacy is required. Furthermore, as emphasized by [31] ([31]), these differences extend into technological self-efficacy, which is increasingly vital in digitally driven fields.

Regarding H3, the analysis did not reveal a statistically significant interaction between gender and academic performance categories for either situational or personal efficacy beliefs. These findings contradict the hypothesis of a combined or moderating effect between these variables. Instead, the results indicate that only gender significantly influences personal efficacy beliefs (H = 15.29, *p* < 0.001, ε^2^ = 0.064), with male students consistently reporting higher levels. In contrast, academic performance (H = 0.28, *p* = 0.598) and its interaction with gender (H = 0.52, *p* = 0.470) were not statistically significant, suggesting that performance level does not meaningfully impact self-efficacy—either independently or in interaction with gender. Similarly, no significant effects were observed for situational efficacy beliefs, which reinforces the idea that students’ perceptions of the academic environment are relatively stable across gender and performance levels. These results underscore the importance of distinguishing between the two self-efficacy components: while personal efficacy appears sensitive to gender-based differences—potentially shaped by internalized beliefs and sociocultural influences—situational efficacy remains largely unaffected by demographic or academic achievement variables. This is consistent with previous research that has emphasized the independent and context-specific nature of gender effects on academic self-efficacy ([38]; [46]), particularly in domains such as business education.

These findings reinforce the idea that academic self-efficacy is a multidimensional and dynamic construct. While personal efficacy beliefs are clearly shaped by gender, they appear to be unaffected by academic performance levels or their interaction. This supports the notion that self-efficacy is not a static trait but one that evolves in response to social and psychological factors ([61]). The results reaffirm Bandura’s theory of self-efficacy ([6]), emphasizing the motivational role of personal beliefs in academic contexts, even in the absence of a direct performance linkage. The lack of correlation between self-efficacy and academic performance echoes prior findings suggesting that this relationship is often mediated by variables such as social support, prior success, and stress ([43]). Furthermore, these findings underscore the need for specific interventions to address gender disparities in personal efficacy beliefs in higher education, especially in disciplines where women face cultural and social barriers. As noted by [57] ([57]), stress, self-imposed demands, and social norms can disproportionately affect women, weakening their personal efficacy beliefs despite their objective performance being equal to or surpassing that of their male counterparts. Specific mentoring programs, empowerment workshops, and activities designed to strengthen personal efficacy beliefs can be effective tools in this regard ([45]; [60]; [62]; [63]). Addressing these disparities requires targeted interventions that enhance personal efficacy beliefs among female students, particularly through mentorship programs, leadership development workshops, and inclusive curricula ([61]). In addition, interventions that reduce women’s perceived barriers and encourage their participation in competitive fields are critical to advancing gender equity ([36]; [38]; [45]). [64] ([64]) and [81] ([81]) assert that a learning environment promoting gender equality and minimizing stereotypes contributes positively to the development of self-efficacy in female students. These findings underscore the relevance of implementing education policies that promote a climate of mutual support and respect among all students, regardless of their gender.

This study has several limitations that warrant consideration. The sample consisted solely of business students from one particular region of Chile, perhaps constraining the generalizability of the findings. In addition, academic performance was self-reported, potentially introducing bias into the data. Another limitation is the homogeneity of the sample in terms of educational and cultural context, which could influence the lack of statistical significance in some findings, particularly in the analysis of situational self-efficacy.

Future research could expand the sample size and integrate a broader range of disciplinary areas to assess the generalizability of the results. Longitudinal designs might be beneficial for examining the progression of self-efficacy during the academic program and its influence on professional success. Employing mixed methodologies would provide a more in-depth exploration of the dynamics associated with gender stereotypes and their impact on academic self-efficacy. Moreover, future studies could investigate the effectiveness of various interventions aimed at improving the personal self-efficacy of female students. Finally, implementing and evaluating experimental programs designed to reduce gender gaps would provide valuable evidence to guide institutional strategies that promote equity and academic success. Addressing these research gaps will contribute to more inclusive educational environments and inform evidence-based strategies to support students’ self-efficacy development.

## 5. Conclusions

This study offers meaningful insights into gender differences in academic self-efficacy among business students, emphasizing the importance of tailored interventions. The findings indicate that women, in particular, could benefit from initiatives that strengthen personal efficacy beliefs and address cultural barriers. Future research should explore these dynamics in greater depth—employing longitudinal and mixed-method designs—and assess specific programs aimed at enhancing personal self-efficacy among female students. This would contribute to a more holistic understanding of self-efficacy and its influence on business education. In this regard, future studies could also consider the impact of educational programs such as mentoring, leadership training, or self-regulated learning workshops, which have shown potential to support self-efficacy development.

## Figures and Tables

**Figure 1 behavsci-15-00563-f001:**
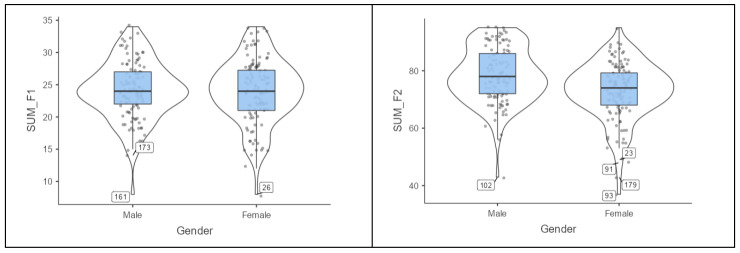
Distribution of academic self-efficacy by gender.

**Table 1 behavsci-15-00563-t001:** Sample characteristics (n = 239).

	General Sample		By Gender			
			Female n = 124		Male n = 115	
	Mean	SD	Mean	SD	Mean	SD
Age	21.60	2.33	21.38	2.23	21.74	2.44
Academic Performance	5.46	0.36	5.56	0.38	5.46	0.34
Level in Curriculum						
2nd year	23.7%		22.58%		25.22%	
3rd year	25.6%		23.39%		27.83%	
4th year	23.9%		27.42%		20.00%	
5th year	26.8%		26.61%		26.96%	

**Table 2 behavsci-15-00563-t002:** Standardized regression weights for all items.

Factors/Item	Estimate	S.E.	A	ω
Factor 1			0.708	0.713
X2	0.430	0.076		
X5	0.475	0.078		
X11	0.711	0.061		
X12	0.569	0.065		
X13	0.575	0.060		
X18	0.472	0.068		
X23	0.325	0.081		
X27	0.542	0.062		
Factor 2			0.92	0.929
X1	0.604	0.040		
X3	0.747	0.032		
X4	0.781	0.031		
X6	0.560	0.043		
X7	0.772	0.031		
X8	0.829	0.025		
X9	0.865	0.021		
X10	0.620	0.042		
X14	0.708	0.038		
X15	0.626	0.041		
X16	0.810	0.024		
X17	0.608	0.040		
X19	0.585	0.039		
X20	0.750	0.031		
X21	0.749	0.033		
X22	0.850	0.025		
X24	0.315	0.062		
X25	0.727	0.034		
X26	0.719	0.036		
X28	0.589	0.041		

Note: α = Cronbach’s alpha; ω = Omega de McDonald. Items are presented as retained in the CFA over the total number of items on the original scale.

**Table 3 behavsci-15-00563-t003:** Item analysis by factor.

Factor/Items	Mean	SD	Skewness	Kurtosis
Factor 1	3.01	0.592		
X2	2.53	0.911	0.440	−0.096
X5	3.26	1.120	−0.137	−0.808
X11	3.69	1.066	−0.559	−0.307
X12	2.59	0.974	0.531	−0.183
X13	2.96	1.076	0.104	−0.639
X18	2.92	1.054	0.173	−0.628
X23	3.57	1.135	−0.409	−0.620
X27	2.52	0.893	0.390	0.046
Factor 2	3.97	0.539		
X1	3.85	1.008	−1.017	0.414
X3	4.06	0.794	−0.874	0.792
X4	4.14	0.699	−0.716	0.985
X6	3.56	1.047	−0.506	−0.558
X7	4.44	0.706	−1.644	41.363
X8	4.16	0.694	−0.686	0.8597
X9	4.10	0.746	−0.846	10.414
X10	4.18	0.720	−0.761	11.862
X14	4.02	0.820	−0.862	0.847
X15	3.86	1.039	−1.047	0.640
X16	4.03	0.767	−0.720	0.894
X17	3.56	1.154	−0.565	−0.622
X19	4.10	0.736	−0.600	0.339
X20	4.05	0.776	−1.011	16.177
X21	3.71	1.006	−0.692	−0.138
X22	3.88	0.780	−0.704	0.774
X24	3.83	3.83	−0.696	0.853
X25	4.12	4.12	−0.807	0.852
X26	4.44	4.44	−0.708	1.038
X28	3.26	3.26	−0.266	−0.762

**Table 4 behavsci-15-00563-t004:** Descriptive statistics of academic self-efficacy factors by performance group.

Factor	GPA Range	n	Mean	SD
F1: Situational efficacy beliefs	<=5.4	116	24.1	5.28
	5.5–5.6	58	23.7	4.13
	>=5.7	65	23.9	4.24
F2: Personal efficacy beliefs	<=5.4	116	76.0	9.76
	5.5–5.6	58	72.8	11.71
	>=5.7	65	77.1	9.17

**Table 5 behavsci-15-00563-t005:** Comparison of gender differences by academic self-efficacy factor.

Factor	Gender	Mean	*p*-Value
F1: Situational efficacy beliefs	Male	24.1	0.919
	Female	24.0	
F2: Personal efficacy beliefs	Male	78.4	<0.001
	Female	72.8	

## Data Availability

The data that support the findings of this study are available from the corresponding author upon reasonable request.

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
