# Peer review of "Gender, Self-Efficacy, and Academic Performance: Evidence in Business Education Program"

_behavsci, 2025, doi:10.3390/bs15050563_

Round 1

Reviewer 1 Report

Comments and Suggestions for Authors

Congratulations to the authors on a well-constructed article with novel and useful findings for business education.

The paper is well-written and contains references appropriate for the research.

Other than some minor editing (such as incorrect use of the capital letter on line 20 for "Personal" and incorrect placement of "2." on line 160, there is little more that is required for this paper. Perhaps the authors could consider future research that investigates the effectiveness of various interventions to improve the personal self-efficacy of female students.  

Author Response

We are grateful for your positive evaluation of the manuscript, especially your recognition of its novel contribution and relevance to business education.

All changes are highlighted in yellow on the corresponding lines.

Reviewer 2 Report

Comments and Suggestions for Authors

I would like to thank you for giving me the opportunity to read this manuscript.
The study is interesting and most of it is well-structured & cohesive.
The theoretical foundation effectively integrates Bandura's self-efficacy theory and numerous relevant studies. Furthermore, gender disparities in self-efficacy, particularly in business education, are highly relevant.
The study is well-supported by literature, but some areas could be more critically analyzed rather than just summarized. The statistical analysis adds credibility, and findings align with prior research. Also it offers recommendations for interventions, mentorship programs, and policy improvements.
But, there is room for improvement. Here are some comments which I believe can contribute to its quality. Some ideas for example: gender stereotypes affecting self-efficacy, are repeated across sections—could be more concise.
The English language can be edited in a better way: some sentences are too long and dense, making readability a challenge. There are several grammar & minor corrections for example:
"Albert Bandura (1986) also emphasized that self-efficacy is influenced not only by individual experiences..."
→ "self-efficacy is influenced not only by individual experiences..."
"Huang (2013) reinforces this viewpoint..."
→ "Huang (2013) reinforces this viewpoint..."
Try to avoid shifts between ideas (e.g., from Factor 2 to H3) could be smoother.
 Conclusion Could be edited in a "smother way", Clarity & Readability.
I hope that this review can improve this manuscript.
All the best.

Comments on the Quality of English Language

The English language can be edited in a beter way: some sentences are too long and dense, making readability a challenge. There are several grammar & minor corrections for example:

"Albert Bandura (1986) also emphasized that self-efficacy is influences not only by individual experiences..."
"self-efficacy is influenced not only by individual experiences..."

"Huang (2013) reinforce this viewpoint..."
"Huang (2013) reinforces this viewpoint..."

Try to avoid shifts between ideas (e.g., from Factor 2 to H3) could be smoother.

 Conclusion Could Be edited in a “smother way”, Clarity & Readability.

I hope that this review can improve this manuscript.

All the best.

Author Response

We appreciate your positive assessment of our theoretical framework, structure, and statistical analysis. Your detailed comments helped us enhance clarity and strengthen key sections of the manuscript.

All changes are highlighted in yellow on the corresponding lines.

Reviewer 3 Report

Comments and Suggestions for Authors

As clearly stated in the first two lines of the article (lines 30-31), self-efficacy is closely linked to academic performance. The article, as a whole, examines gender differences. Providing more information about the rationale behind this study could enhance the significance of this article. The gap in understanding (line 143) is a reason, but undoubtedly not the only one.  In the conclusion, the need for customised intervention is mentioned. Some background information on such interventions, specifically the education program, could also be included in the introduction to better understand the rationale for this study. 

The article focuses on the business education program, so many more details on the specifics of this program could raise the importance of this article for readers. 

The conclusion that women (in the business educational program in Chile) may benefit from strengthening their personal efficiency and addressing cultural barriers is well supported by this research. 

Author Response

Thank you for highlighting the importance of strengthening the rationale and context of the study. Your comments were key to refining the introduction and clarifying the significance of the research.

All changes are highlighted in yellow on the corresponding lines.
